**Title: Climate change and the global pattern of moraine-dammed glacial lake outburst floods**

**Authors:**

Stephan Harrison*[1], Jeffrey S. Kargel[2], Christian Huggel[3], John Reynolds[4], Dan H. Shugar[5], Richard A Betts[1, 6], Adam Emmer[7,10], Neil Glasser[8], Umesh K. Haritashya[9], Jan Klimeš [,11], Liam Reinhardt[1], Yvonne Schaub[3], Andy Wiltshire[6], Dhananjay Regmi[12], Vít Vilímek[7]

**Affiliations:**

1. College of Life and Environmental Sciences, Exeter University, U.K.

2. Planetary Science Institute, Tucson, AZ 85719, USA; and Department of Hydrology & Atmospheric Science, University of Arizona, Tucson, AZ 85742, USA.

3. Department of Geography, University of Zurich, CH-8057 Zurich, Switzerland

4. Reynolds International Ltd, Suite 2, Broncoed House, Broncoed Business Park, Wrexham Road, Mold, Flintshire, UK.

5. Water, Sediment, Hazards, and Earth-surface Dynamics Laboratory, University of Washington Tacoma, WA, 98402

6. Met Office Hadley Centre, FitzRoy Road,Exeter Devon U.K.

7. Department of Physical Geography and Geoecology, Charles University in Prague, Faculty of Science, Albertov 6, 128 43 Praha, Czech Republic

8.  Centre for Glaciology, Department of Geography and Earth Sciences, Aberystwyth University, Wales SY23 3DB, U.K.

9.  Department of Geology, University of Dayton, 300 College Park, Dayton, OH 45469-2364

10. Department of Human Dimensions of Global Change, Global Change Research Institute, Czech Academy of Sciences, Bělidla 986/4a, 60300 Brno, Czech Republic.

11. Department of Engineering Geology, Institute of Rock Structure and Mechanics, Czech Academy of Sciences, V Holešovičkách 41, 182 09 Prague 8, Czech Republic.


12. Himalayan Research Center, Lainchaur, Kathmandu, Nepal


**Corresponding author**: Stephan Harrison, College of Life and Environmental Sciences, Exeter University, Cornwall Campus, TR10 9EZ, U.K.


**Keywords**:  Climate change, GLOF, hazards, jökulhlaup, time series, moraine-dammed lake






**Abstract:**

Despite recent research identifying a clear anthropogenic impact on glacier recession, the effect of
recent climate change on glacier-related hazards is at present unclear.  Here we present the first global
spatio-temporal assessment of glacial lake outburst floods (GLOFs) focusing explicitly on lake drainage
following moraine dam failure.  These floods occur as mountain glaciers recede and downwaste. GLOFs
can have an enormous impact on downstream communities and infrastructure.  Our assessment of
GLOFs associated with the rapid drainage of moraine-dammed lakes provides insights into the historical
trends of GLOFs and their distributions under current and future global climate change.  We observe a
clear global increase in GLOF frequency and their regularity around 1930, which likely represents a
lagged response to post-Little Ice Age warming.  Notably, we also show that GLOF frequency and their
regularity —rather unexpectedly—has declined in recent decades even during a time of rapid glacier
recession. Although previous studies have suggested that GLOFs will increase in response to climate
warming and glacier recession, our global results demonstrate that this has not yet clearly happened.
From assessment of the timing of climate forcing, lag times in glacier recession, lake formation and
moraine-dam failure, we predict increased GLOF frequencies during the next decades and into the 22[nd]
century.

## 1.  Introduction

There is increasing scientific and policy interest in detecting climate change impacts and assessing the
extent to which these can be attributable to anthropogenic or natural causes.  As a result, recent
research demonstrating an anthropogenic fingerprint on a significant proportion of recent global glacier
recession is an important step forward (Marzeion et al. 2014).  The focus can now shift to glacier hazards
but the complex nature of glacier-climate interactions (Roe et al. 2017)  and their influence on hazards
makes this a challenging task (Shugar et al. 2017).
Mountain glaciers have continued to recede (Kargel et al. 2014; Cramer et al. 2014) and thin from their
late Holocene (Little Ice Age; LIA) positions and, in many cases, the rate of recession and thinning has
increased over recent decades largely as a consequence of global warming (Marzeion et al. 2014).
Thinning, flow stagnation and recession of glacier tongues have resulted in formation of moraine-
dammed lakes (Richardson and Reynolds 2000).  These moraines, some of which contain a melting ice
core, are built from rock debris transported by glaciers. When they fail, large volumes of stored water
can be released, producing glacial lake outburst floods (GLOFs).  These floods have caused thousands of
fatalities and  severe impacts on downstream communities, infrastructure and long-term economic
development (Mool et al. 2011; Riaz et al. 2014; Carrivick and Tweed 2016).

Although much research has been carried out on the nature and characteristics of GLOFs and hazardous lakes from many of the world's mountain regions (e.g. Lliboutry et al. 1977; Evans 1987; O'Connor et al. 2001; Huggel et al. 2002; Bajracharya and Mool 2009; Ives et al. 2010; Iribarren et al. 2014; Lamsal et al. 2014; Vilimek et al. 2014; Westoby et al 2014; Perov et al 2017), there are significant gaps in our knowledge of these phenomena at the global scale and concerning their relationship to anthropogenic climate change. Detecting changes in the magnitude, timing and frequency of glacier-related hazards over time and assessing whether changes can be related to climate forcing and glacier dynamical responses is also of considerable scientific and economic interest (Oerlemans 2005; Stone et al. 2013). Multiple case studies are insufficient to achieve a better understanding of the mechanisms leading to GLOF initiation so a more comprehensive understanding of the global frequency and timing of GLOFs is necessary. Testing such relationships at a global scale is also an important step toward assessment of the sensitivity of geomorphological systems to climate change.

Despite numerous inventories of GLOFs at regional scales (see Emmer et al 2016), no global database has been created which focuses specifically on GLOFs relating to the failure of moraine dams. A global database is required to place GLOFs in their wider climatic context (Richardson and Reynolds 2000; Mool et al. 2011). This means that we are unable to answer some important questions concerning their historic behaviour and therefore the changing magnitude and frequency of GLOFs globally through time, and their likely evolution under future global climate change. This latter point is made even more difficult by the lack of long-term climate data from many mountain regions. Given the size and impacts of GLOFs in many mountain regions, better understanding their links to present and future climate change is of great interest to national and regional governments, infrastructure developers and other stakeholders. We argue that glacier hazard research needs to be increasingly seen through the lens of change adaptation.

These issues and knowledge gaps can be addressed via a systematic, uniform database of GLOFs. Here we have compiled an unprecedented global GLOF inventory related to the failure of moraine dams. We discuss the problems involved in developing a robust attribution argument concerning GLOFs and climate change. This inventory covers only the subset of GLOFs that are linked to overtopping or failure of moraine dams. Our focus on moraine dams is motivated by: 1) this type of event leaves clear diagnostic evidence of moraine-dam failures in the form of breached end moraines and lake basins, whereas ice-dammed lake failures commonly do not leave such clear and lasting geomorphological

evidence; and 2)  the conventional hypothetical link between climate change, glacier response, moraine-
dammed lake formation and GLOF production is more straightforward compared to the range of
processes driving GLOFs from ice- and bedrock-dammed lakes.
Such GLOF events are often triggered by ice and rock falls, rock slides or moraine failures into lakes
creating seiche or displacement waves, but also by heavy precipitation or ice/snow melt events
(Richardson and Reynolds 2000).  While climate change plays a dominant role in the recession of
glaciers, downwasting glacier surfaces debuttress valley rock walls leading to catastrophic failure in the
form of rock avalanches or other types of landslides (Ballantyne 2002; Shugar and Clague 2011; Vilimek
et al. 2014).  Other climatically induced triggers of moraine dam failures include increased permafrost
and glacier temperatures leading to failure of ice and rock masses into lakes and the melting of ice cores
in moraine dams which leads to moraine failure and lake drainage.
Attribution of climate change impacts is an emerging research field and no attribution studies on GLOFs
are available so far. Even for glaciers only very few attribution studies have been published to date
(Marzeion et al. 2014; Roe et al. 2017). Follow-up studies from the IPCC 5[th] Assessment Report (Cramer
et al. 2014) proposed a methodological procedure to attribute impacts to climate change (Stone et al.
2013).  Based on that, a methodologically sound detection and attribution study needs first to formulate
a hypothesis of potential impact of climate change. In our case physical process understanding supports
the association between climate change and GLOFs associated with moraine-dam failure by climate
warming resulting in glacier recession and glacial lake formation and evolution behind moraine dams
which become unstable and fail catastrophically. The next step requires a climate trend to be detected,
followed by the identification of the baseline behaviour of the system in the absence of climate change.
The difficulty of identifying the baseline behaviour is related to several factors.  The first is the existence
of confounding factors, both natural and human related. For instance, the frequency of GLOFs from
moraine dams also depends on factors such as the stability of the dam, including dam geometry and
material, or mitigation measures such as artificial lowering of the lake level (Portocarreo-Rodriguez
2014). Second, there are few long-term palaeo-GLOF records with which to assess baseline behaviour.
Eventually, attribution includes the detection of an observed change that is consistent with the response
to the climate trend, in our case a change in GLOF occurrence, and the evaluation of the contribution of
climate change to the observed change in relation to confounding factors. Our chief observational result
is that there is an upsurge in GLOF frequency starting around 1930 and then a decline following roughly
1975 and persisting for decades (see also Carrivick and Tweed 2014). At face value, when comparing
with the climate records, there seems to be no relationship between global GLOF frequency and
concurrent climatic fluctuations, and a regional breakdown offers no solution; for example, strong
climatic global (or Northern Hemisphere) warming during the period of declining GLOF frequency after
1975 appear to be inconsistent. A simplistic inference would be that climate change does not influence
GLOF incidence, but we reject this given our understanding of the physical drivers of glacier recession,
lake development and drainage mechanisms.   Although we know that GLOFs involve a complex set of
dynamics, one of the important dynamical changes affecting GLOFs is the formation and growth of
glacial lakes, and we know that there must be a relationship here to climatic warming.  GLOF triggers
also commonly involve extreme weather, such as extreme heat and extreme precipitation, which are
intuitively linked to climate change as well, even if the attribution experiments have not yet been
carried out. We thus have to dig deeper to see how GLOF frequency may be connected to climate
change.  The point arises that the conditions needed for a GLOF involve a long period of lake formation
and growth, such that past climate changes are involved.  In the Methods section we produce a model
whereby the history of one climate variable and its time derivative-- Northern Hemisphere mean
temperature and warming rate-- are linked to the GLOF record.

## 2.  Methods

We produced a database of  GLOFs developed from a collation of regional inventories and reviews (e.g.
GAPHAZ, WGMS and GLACIORISK databases and the GLOF Database provided under ICL database of
glacier and permafrost disasters from the University of Oslo ) and regional overviews and reviews (e.g.
Clague et al 1985; Xu 1987; Costa and Schuster 1988; Reynolds 1992; Ding and Liu 1992; Clague and
Evans 2000; O'Connor et al 2001; Zapata 2002; Raymond et al 2003; Jiang et al 2004; Carey 2005; Osti
and Egashira 2009; Narama et al 2010; Ives et al 2010; Wang et al 2011; Carey et al 2011; Mergili and
Schneider 2011; Fujita et al 2012;Iribarren et al 2014 and Emmer et al 2017, and case studies of
individual GLOFs (eg Kershaw et al 2005; Harrison et 2006; Worni et al 2012).  A complete list is available
in the **Supplementary Information File**). The GLOF database was developed from a collation of regional
inventories and reviews (**Supplementary Information File**).  Only GLOFs that could be dated to the year
and to moraine failure were included. Past temperature trends from the glacier regions of interest were
extracted from three independent global temperature reconstructions (CRUTEM4.2 (Jones et al. 2012),
NOAA NCDC (Smith et al. 2008) and NASA GISTEMP (Hansen et al. 2010). These datasets provided
temperature anomaly data relative to a modern baseline beginning in 1850 for CRUTEM4.2 and 1880 for
NOAA NCDC and NASA GISTEMP.
**2.1  Test of direct linkage between GLOF rate and climate change**
We concentrate exclusively on the subset of GLOFs associated with the failure of moraine-dammed
lakes as these are a major hazard in many mountain regions but also represent the best candidates of
outburst floods for attribution to climate change.  We differentiate these from other glacially sourced
outburst floods, such as those resulting from the failure of an ice dam (Walder and Costa 1996; Tweed
and Russell 1999; Roberts et al. 2003), dam overflow; volcanically triggered jökulhlaups (Carrivick et al.
2004; Russell et al. 2010; Dunning et al. 2013) or the sudden release of water from englacial or
subglacial reservoirs (Korup and Tweed 2007).
The period over which climate data are available is dependent on the region but starts in 1850 in
CRUTEM4.2 and 1880 in NOAA NCDC and NASA GISTEMP. The resolution of the data is generally 5
degrees; however, NASA GISTEMP is provided at 1 degree resolution but it should be noted this does
not imply there are more observational data in this analysis. For each region, we extract all gridpoints
that contain a glacier as defined in the Extended World Glacier Inventory (WGI-XF). With the exception
of the European Alps no dataset contains a complete continuous record for the period 1900-2012. We
therefore take all available datapoints to form time series for each dataset and derive a mean linear
trend for the 1990-2012 period. Given large uncertainties and data gaps no attempt is made to
statistically test these trends.  The trends presented here are therefore considered illustrative of past
changes in temperature for these regions.
**2.1.1 Wavelet analysis of GLOF incidence**
Wavelets are a commonly used tool for analyzing non-stationary time series because they allow the
signal to be decomposed into both time and frequency (e.g. Lane 2007). Here, we follow the
methodology of Shugar et al. (2010), although we use the Daubechies (db1) continuous wavelet. The
wavelet power shown here have been tested for significance at 95% confidence limits, and a cone of
influence applied to reduce edge effects. We follow Lane (2007), in choosing an appropriate number of
scales (S=28, see his eqn 28), which is related to the shape of the cone of influence.

**2.2  The Earth's recent climate record smoothed along glacier response timescales: development of**
**the GLOF lag hypothesis**
A potentially destructive GLOF may elapse after a glacial lake  grows to a volume where sudden release
of glacial lake water can exceed a normal year's peak instantaneous discharge.  There are time scales
associated with the period between a climatic (or other) perturbation and the occurrence of a GLOF. The
following thought experiment demonstrates the concept of the lagging responses of GLOF activity to
climate change: an initialized stable condition allows glacier-climate equilibrium, where neither climate
nor glacier has fluctuated much for some lengthy period, and where no other strongly perturbing
conditions exist, e.g., there are no significant supraglacial or ice-marginal or moraine-dammed lakes, and
a steady state exists in the supply and removal of surface debris.  We then impose a perturbation
(climatic or other) which favours eventual lake development and growth and eventually a GLOF. We
describe two successive time periods which must pass before a significant GLOF can occur, and then a
third period before a GLOF actually occurs: lake-inception time ($\tau_i$), lake growth time ($\tau_g$), and trigger
time ($\tau_t$).  The first two sum to the GLOF response time ($\tau_{GLOF}$); as we define it, $\tau_{GLOF} = \tau_i + \tau_g$. The terms
are for illustrative purposes; many supraglacial ponds initially go through a lengthy period where they
fluctuate and drain annually and thus do not have a chance to grow beyond one season.  Furthermore,
lakes can grow to a point where limnological processes take over from climate, hence lake growth
becomes detached from climate change.  Even so, our set of definitions can be used to explain the
lagging responses of glacier lakes and GLOFs to climatic history.

A GLOF does not necessarily occur upon climate step change date + $\tau_{GLOF}$, which is the timescale over
which the metastable system establishes a condition where a significant GLOF *could* occur. A trigger is
needed (e.g., a large ice or rock avalanche into the lake or a moraine collapse as an ice core melts).
After a sizeable glacial lake has developed, suitable GLOF triggers may occur with a typical random
interval averaging $\tau_t$, which depends  on the topographic setting of the glacier lake, valley-side geology,
steepness, moraine dam properties and climate.  As a result, $\tau_t$ could range from years to centuries.
Furthermore, as a lake usually continues to grow after $\tau_{GLOF}$ has elapsed, $\tau_t$ can in principle change,
probably shortening as the lake lengthens and as the damming moraine degrades. The time elapsing
between a climatic perturbation and a GLOF then is the sum of three characteristic  sequential periods,
$\tau_i + \tau_g + \tau_t$.

The lake inception time $\tau_i$ might be approximated by the glacier response time, which has been defined
parametrically (Johanneson et al. 1989; Bahr et al. 1998) but in general describes a period of adjustment
toward a new equilibrium following a perturbation.  We take a simple parameterization (Johanneson et
al. 1989) and equate $\tau_l = h/b$, where $h$ is the glacier thickness of the tongue near the terminus and $b$ is
the annual balance rate magnitude. The glacier response time approximating the lake inception time
may be many decades for most temperate valley glaciers, but it can range between a few years and a
few centuries.  The glacier response time is  a climate-change forgetting timescale. After a few response
times have elapsed, a glacier's state and dynamics no longer remember the climate change that induced
the response to a new equilibrium. For illustration, we adopt $\tau_l$ = 60 years, a value typical of many
temperate valley glaciers .

A supraglacial pond  may drain and redevelop annually (posing no significant GLOF risk), but at some
point, if there is a sustained long-term negative mass balance, supraglacial ponds commonly grow,
coalesce and form a water body big enough that rapid partial drainage can result in a significant GLOF.
That lake growth period is defined here as $\tau_g$, for which we adopt 20 years, a value typical of many
temperate glacier lakes of the 20[th] century (e.g. Wilson et al., 2018; Emmer et al. 2015)  Hence, $\tau_{GLOF} = \tau_i$
$+ \tau_g \approx$ 80 years for the favoured values.  Hence, a significant GLOF may occur at any time from 80 years
following a large climatic perturbation; what the GLOF waits on is $\tau_t$ , which could be years or a  century.
This concept can be extended to the lagging response of a whole population of glaciers following a
perturbation in regional climate (Fig. 1).

We distinguish between climate change, which may establish conditions needed for a GLOF to happen,
and weather, which sometimes may be involved in a GLOF trigger.  GLOF triggers are diverse, e.g.,
protracted warm summer weather may trigger an ice avalanche into the lake or moraine melt-through,
or heavy winter snow may trigger an ice avalanche into the lake.
However, the relevant controlling climate, in this example, is that of the prior climatic history and the
conditioning period defined by $\tau_{GLOF}$ and the typical trigger interval $\tau_t$. Hence, $\tau_{GLOF}$  is closely connected
to climate, whereas $\tau_t$ can be connected to weather for certain types of triggers.

The assessment above is for a single step-function climate change.  Considering that climate changes
continuously and glacier characteristics vary, populations of glaciers must have full distributions of $\tau_i$, $\tau_g$,
and $\tau_{GLOF}$. Even while glaciers are still adjusting to any big recent historical climate change, more climate
change accrues; glacier and lake dynamics take all that into account, either increasing the likelihood and
perhaps size of a GLOF or decreasing or delaying it. Hence, the overall GLOF frequency record cannot be
synchronous with climatic fluctuations, and it also should not simply trace past climate change with a
time lag; rather, the GLOF frequency record for any large population of glaciers should be definitely but
complexly related to the recent climatic history.

The functional dependence on climate history is not known for any glacier or population of glaciers, but
to explore the concept of a lagged GLOF response to accrued climate changes, we assert that the
integration function will tend to weight recent climatic shifts more strongly than progressively older
climatic shifts, the memory of which is gradually lost as the glacier population adjusts.  That is, because
of glacier dynamics and the responses of a population of glaciers to climatic changes, the population
eventually loses memory of sufficiently older climatic changes and adjusts asymptotically toward a new
equilibrium. This should be true for any climate-sensitive glacier dynamics (Oerlemans 2005). Though we
do not know the functional form of the glacier responses (either for an individual glacier or a
population), we nonetheless wish to illustrate our point while not driving fully quantitative conclusions.
We propose that the integration of climate information into ongoing glacier dynamical adjustments
occurs with exponentially declining weighting going backward in time from any given year.  The
exponential time weighting constant may be similar to $\tau_{GLOF}$ . We have computed a moving time-average
northern hemisphere temperature with the weighting of the average specified by an assumed $\tau_{GLOF}$ = 80
years; the computed moving average pulls data, for any year, over the preceding period of $\tau_{GLOF}$ , i.e.,
includes temperature information up to 240 years prior to any given year. The weighting of earlier years'
temperatures within that $\tau_{GLOF}$  is less than that of later years, according to the exponential. The cutoff at
$\tau_{GLOF}$  is arbitrary, and was done for computational expediency, seeing that any climate fluctuation
occurring before $\tau_{GLOF}$  years earlier is inconsequential due to the exponential memory loss.
We  combined the Mann et al. (2008 ) multi-proxy Northern Hemisphere temperature anomaly from
501 AD to 1849, the Jones et al. (2012) (https://crudata.uea.ac.uk/cru/data/temperature/#datdow)
Northern Hemisphere land instrumental temperature record from 1850 to 2014, and a model of
expected warming from 2015 to 2100. It is the recent climate history at each glacier lake or region that
is strictly relevant, but lacking such records, and needing here to only establish the concept, we settle
for the treatment described above involving the Northern Hemisphere temperature anomaly.

The model is a constant 2.7 **°**C/century warming; noise was added from a naturally noisy but overall
non-trending instrumental record from 1850 to 1899, with some years repeated, to append the 2015-
2100 period (Fig 1).  The Mann et al.(2008) and Jones et al. (2012) Datasets were brought into
congruence in 1850. Then we smoothed the composite record + model results using the $\tau_{GLOF}$
exponentially weighted filter, as described above, where the natural logarithmic "forgetting" timescale
$\tau_{GLOF}$ = 20, 40, or 80 years for three illustrative cases. Smoothing was computed for $\tau_{GLOF}$ , i.e., 240 years
if $\tau_{GLOF}$ = 80 years. Our favoured value $\tau_{GLOF}$ = 80 years is based on large Himalayan and other temperate
glacier lakes.  The shorter response times would likely apply to small  glaciers, or those occuring in steep
valleys.

Regardless of the functional form of the glacier response and lake dynamics, GLOF frequency in any
given region or worldwide must  lag the climate record.  The historically filtered/smoothed temperature
record + model incorporating $\tau_{GLOF}$ = 20, 40, and 80 years is shown in Fig 1A though C together with the
unsmoothed actual record + model temperature series. The temperature anomalies are plotted in
panels A, B, and C; and the warming rate in panels D and E.  The historically averaged/smoothed
temperature record lags  fluctuations in the unsmoothed record.  The lag is most easily seen where
temperatures start to rise rapidly in the 20[th] and 21[st] centuries.  The high-frequency temperature
anomaly fluctuations also show concordantly but in damped form in the smoothed moving average
curves because the curves are historical moving averages with heaviest weighting toward the more
recent years.  The lagging responses are also seen at several times when the running average curves
variously show warming and cooling for the same year depending on the value of $\tau_{GLOF}$ .

We posit that the historically filtered warming rate (more than the temperature anomaly) drives GLOF
frequency.  In Fig 1 we show GLOF frequency (smoothed over 10-year moving averages) together with
the warming rate extracted from the historically filtered temperature + model temperature time series.
To get a better match with the temperature treated as such, we applied a further 45-year shift. From a
glacier and lake dynamics perspective, this shift might relate to the trigger time scale, $\tau_t$. Singular values
of $\tau_{GLOF}$ and $\tau_t$ should not pertain globally to all glaciers; but should span wide ranges. The adopted values
$\tau_{GLOF}$ = 80 years and $\tau_t$ = 45 years nonetheless make for a plausible match between the GLOF and
climate records.   These numbers make sense in terms of glacier and lake dynamics timescales, but we
reiterate that our purpose with this climate-GLOF fitting exercise is illustrative. In sum, a notable shift in
GLOF frequency does not connote a concordant shift in climate, though prior climate change may still
underlie the cause.


## 3. Results

Our global analysis identifies 165 moraine-dam GLOFs, recorded since the beginning of the 19[th] century (Fig. 2A). The vast majority of these GLOFs (n=160; 97%) occurred since the beginning of the 20[th] century, at a time of climate warming and increasing glacier recession (Fig. 2 and 5). None of these GLOFs were associated with repeat events from the same lake. Around 65% of GLOFs occurred between 1930 and 1990. Thirty-six GLOFs occurred in the mountains of western North America between 1929 and 2002 (SI Table 1). Fifteen of these occurred in western Canada, 15 in the Cascades Range of the US and four in Alaska. One occurred in Mexico and 1 in the Sierra Nevada. In the South American Andes we identified 40 GLOFs. Eleven occurred in Chile between 1913 and 2009 (including the large one in Patagonia at Laguna del Cerro Largo in 1989); one in Colombia in 1995 and 28 in Peru between 1702 and 1998. Fourteen GLOFs are listed from the European Alps. Three are from Austria between 1890 and 1940; five from Switzerland between 1958 and 1993; one from France in 1944 and five from Italy between 1870 and 1993. In the Pamir and Tien Shan mountains in central Asia, we identified 20 GLOFs, with most of these dating from the late 1960s to the early 1980s. The largest number of GLOFs (55) is reported from the Hindu Kush Himalaya (HKH) including the mountains of Bhutan and Tibet, dated from the 20[th] and 21[st] century. Thirty are from Tibet (between 1902-2009); 12 from Nepal between 1964 and 2011(and one is reported to have occurred in 1543), and five in Pakistan between 1878 and 1974. There is uncertainty in reporting some of these GLOFs and we discuss this further in the Supplementary Information File.

Starting around 1930 until about 1950, GLOFs occurred with regularity but a low frequency (Fig. 3). In other words, floods occurred with relatively long period variability (50-60 years). Starting around 1960, the frequency of these events increased (period decreased to approximately 20 years), remaining relatively high until about 1975, after which the statistically significant periodicities end, though GLOFs continue to occur.

While incomplete data restricts a full analysis of GLOF triggers, precise date, magnitude and initiation at a global scale, many GLOFS triggered by ice avalanches and rock falls occur during summer (see Fig. 4). The characteristics of GLOFs that could be influenced by climate change include: changes in magnitude, frequency, timing (either changes in seasonality or changes over longer timescales) and trigger mechanisms. In addition, many rock avalanches into lakes triggering a GLOF may represent a paraglacial response to deglaciation from the LIA or earlier times (Knight and Harrison 2013; Schaub et al. 2013) and

this delayed response demonstrates the need to account for lags between changes in forcing and
responses in attribution studies.

**4.  Discussion**
From this analysis, we highlight three key observations: (1) GLOFs became more common around 1930
but then their incidence was maintained at a quasi-steady level for a few decades thereafter; (2) since
about 1975, GLOF periodicity has decreased globally; and (3) the periodicities of GLOF occurrence has
changed throughout the 20$^{th}$ century. These observations are discussed below.
Our first main observation is that GLOF frequency increased dramatically and significantly around 1930
globally and between 1930 and 1960 regionally (Figs. 1, 2). We find no obvious reason for an abrupt
improvement of GLOF reporting in 1930. While acknowledging that incompleteness of the record must
be a pervasive factor throughout the early period covered by the database we discount reporting
variations as the cause of the abrupt  shift. For instance, this pattern is observed in the European Alps; a
region with a long history of mountaineering, glacier research and valley-floor habitation and
infrastructure development.  Given that we record individual GLOFs in the 19$^{th}$ and early 20$^{th}$ centuries
we argue that the increase in GLOF frequency in the 1930s represents a real increase rather than an
observational artefact.  Following the increase around 1930, we observe a similar rate of GLOFs for the
subsequent years, typically 1 per year in the following decade, increasing to 2-3 per year during the
1940s (e.g. Fig 1A, 2A).  Again, there is no evidence that incompleteness of data is a main cause of the
observed pattern.  We therefore conclude that the incidence of global GLOFs has remained generally
constant between about 1940 and about 1960. In the 1960s and early 1970s, several years saw more
than 5 GLOFs. We argue below that the trend between 1940-60 hides a more complex spatial and
temporal pattern (Clague and Evans 2000; Schneider et al. 2014).
Our second main observation is that while there is considerable variability between regions, GLOF
incidence rates have decreased since about 1975 globally (Fig 2).  There are both more and larger GLOFs
during the 1970s and early 1980s in the Pamir and Tien Shan, in the 1960s in the HKH, and 1990s in
Alaska, the Coast Mountains and Canadian Rockies; and then decreases in both magnitude and
frequency following these periods.  In the Andes however, GLOF incidence decreased after the early
1950s.  The latter observation may be at least partly attributable to considerable GLOF mitigation
measures in Peru, such as engineering based lake drainage or dam stabilization (Carey et al. 2012;
Portocarreo-Rodriguez 2014).  Carrivick and Tweed (2014)  propose  several reasons why 'glacial floods'
may have decreased in frequency in recent decades.  These include successful efforts to stabilize
moraine dams and changes in the ability of fluvial systems to transmit floods over time.  We argue,
conversely, that this reduction may represent a 'lagged' response to glacier perturbations following a
climate change.  More research is clearly needed on this question, and we believe that our analysis,
along with that of Carrivick and Tweed's, will stimulate further work and discussion.

Our third main observation is that for several decades in the 20[th] century, GLOF occurrence has been
periodic, but that periodicity has varied. Since about 1975, and especially since 1990, the periodic nature
of GLOF occurrence has diminished, even though GLOFs have continued. In other words, GLOFs since
1975 have become more irregular. We suspect that the switch to less-periodic outburst floods in recent
decades is related to an underlying mechanism such as topographic constraints and glacier
hypsometries with glaciers retreating into steeper slopes, implying a reduced rate of  moraine-dammed
lake formation - a phenomenon observed e.g., in the European Alps (Emmer et al., 2015).

The statistics of small numbers affects these regional, time-resolved records, but the overall validity of a
similar mid-20[th] century increase and then decrease in the frequency of GLOFs can be further detected
in the global record and is statistically significant (Fig 3). We argue that the reduction in global GLOF
frequency after the 1970s (especially in Central Asia, HKH and North America) is real, because the
contemporary reporting is likely to be nearly complete given the scientific and policy interest in glacier
hazards from the late-20[th] century.  Hence, our conclusion is that globally and regionally there have
been inter-decadal variations in the frequency of GLOFs, and in general the most recent couple of
decades have seen fewer GLOFs than during the early 1950s to early 1990s.  The record's
(in)completeness is not able to explain a decreasing incidence rate.  This temporal variation of GLOF
frequency, and recent decrease, is therefore a robust and surprising result and has occurred despite the
clear trend of continued glacier recession and glacier lake development in recent decades.
Our data allow us to test and refine the widespread assumption that GLOFs are a consequence of recent
climate change (Bajracharya and Mool 2011; Riaz et al. 2014).  This is an important assumption because
it implies that GLOF frequency will increase as the global climate continues to warm with potential
major impacts for downstream regions.
The global increase in GLOF frequency after 1930 must be a response to a global forcing, considering
global glacier retreat (Zemp et al. 2015), and physical process understanding suggests that this is a
lagged response to the warming marking the end of the LIA (Clague and Evans 2000).  Although the
global response appears sudden, in 1930, the region-by-region assessment shows that the response was
asynchronous regionally and temporally over a three decades (Fig 2).  This is consistent with the fact
that the end of the LIA was not globally synchronous (Mann et al. 2009) and also we argue that this
reflects regional variations in glacier response times.
We argue that as a climate shift occurs, after some period related to the glacier response time
previously stable or advancing glaciers start to thin and recede; after a further *limnological response*
*time* proglacial ponds start to grow, coalesce, and deepen into substantial moraine-dammed lakes.
GLOFs typically occur after some additional period of time (the *GLOF response time scale*), but this time
can be brief in glaciers with short response times, such as in the tropical Andes (Fig 1).
In the HKH and central Asia the near-concordant formation of many Himalayan glacier lakes and the
abrupt increase in GLOF rates in the 1950s and 1960s suggests that the GLOF response time is much less
than the limnological response time.  The moraine evidence here indicates that a shift from mainly
glacier advance to recession and/or thinning occurred widely, though regionally asynchronously,
between 1860-1910. The HKH underwent this shift by around 1860 (Owen 2009; Solomina et al 2015) in
response to warming following the regional LIA.  The limnological response time in the Himalayan-
Karakoram region thus is around 100 years, i.e., substantially longer than in the tropical Andes.
We have arrived at a plausible explanation for the post-1930 (1930 to 1960) increases in GLOF rates.
They are most likely heterogeneous, lagging responses to the termination of the LIA, with limnological
response times of the order of decades to 100 years, depending on region (e.g. Emmer et al. 2015). The
limnological response times may be of a similar order to the glacier dynamical response times
(Johanneson et al. 1989; Raper and Braithwaite 2009) but are appended to them.  Thus, measured from
a climatic shift to increased GLOFs, the combined glaciological and limnological response times (plus
GLOF response times, which may be the shortest of the three response times) may sum to roughly 45-
200 years (Fig 1). It cannot be much more than this, because then we would not see the multi-decadal
oscillations in GLOF rates in some regions or globally.
Some individual glaciers may have faster response times than estimated above (Roe et al. 2017), but
taken on a broader statistical basis we infer that most recent GLOFs are a delayed response to the end
of the LIA.  A fundamental implication is that anthropogenic climatic warming to date will likely manifest
in increasing GLOFs in some regions of the world starting early this century and continuing into the 22$^{nd}$
century.  In all the mountain regions considered here the available evidence indicates a warming trend
over the last century around 0.1 °C per decade (Figs 2 and 5). The trend varies between dataset and
region, with the highest rates in the Pamir Tien Shan region and the lowest in the HKH. The most
uncertain region is the Andes, where the sparseness of data prevents any meaningful assessment.  The
trends are consistent with the global mean land temperature trend 0.95±0.02 to 0.11±0.02 °C for 1901-
2012, implying these regions have warmed at approximately the same rate as the global land surface.
The baseline behaviour of glacial lake systems in the absence of climate change is not known in detail,
but the low rate of GLOFs prior to 1930 may indicate that without warming the frequency would be low.
The difficulty of attributing individual GLOF behaviour to climate change relates to the presence of non-
climatic factors affecting GLOF behaviour, such as moraine dam geometry and sedimentology,  climate-
independent GLOF triggers (e.g., earthquakes) and the timescales related to destabilization of mountain
slopes producing mass movements into lakes (Haeberli et al 2016). This represents the period of
paraglaciation (e.g. Ballantyne 2002; Holm et al. 2004; Knight and Harrison 2013).  These system
characteristics may vary regionally and temporally within the evolutionary stage of a receding mountain
glacier, and non-climatic factors such as lake mitigation measures additionally influence GLOF frequency
and magnitude (Clague and Evans 2000; Portocarreo-Rodriguez 2014) .  We argue  that while the
original driver of lake development is likely to involve climate change (resulting in glacier downwasting
and slowed  meltwater flux through glaciers systems as glacier surfaces reduce in gradient) other
mechanical and thermodynamic processes likely assume more importance as the lakes evolve and these
includes small-scale calving and insolation-induced melting of ice cliffs (e.g. Watson et al., 2017).
We also recognise that contemporary mountain glaciers are dissimilar to those that existed in the LIA.
They are, in the main, shorter, thinner and with prominent moraines.  Assumptions that climate
processes acted on similar glacial systems over time are therefore likely to be simplistic.

Based on the analysis of our global GLOF database we have shown that a clear trend is detectable
globally and regionally diversified in the 20$^{th}$ century with a sharp increase of GLOF occurrence around
1930. This trend is attributable to the observed climate trend, namely the warming since the end of the
LIA. The delayed response of GLOF occurrence is an exemplar for the complexities of how natural
systems respond to climate change, underlining the challenges of attribution of climate change impacts.
We have shown here that attribution of GLOFs to climate change is possible although the suite of factors
influencing GLOF occurrence cannot be fully quantified.

In addition, lake outbursts following moraine failures are likely to  be quite different in different regions.
This reflects differences in a number of factors including  ground thermal conditions, presence or
absence of  of ground ice and permafrost, influence of  extreme weather and seismic processes;
topography; and glacial history. To assess these we would need to better understand the
geomorphological time scales involved in lake evolution and failure to design a more robust statistical
analysis and to understand each region's GLOF history.  We thus recommend close attention by the
Earth surface process science community to various process time scales using field studies, satellite
remote sensing, and theoretical modeling.

Our inventory and the global pattern of GLOFs that is derived from it lacks in many cases precise data on
the processes responsible for GLOFs.  This is a consequence of incomplete reporting of GLOFs in remote
mountain regions, especially before the advent and wide use of remote sensing.  In many cases the
record is of a large flood being observed and then some time afterwards a collapsed moraine dam is
seen and the flood attributed to this collapse.  Clearly the precise details of how the collapse occurred is
not always available, and this uncertainty bedevils all similar Detection and Attribution studies,
especially on those events associated with rapid geomorphological change.  This intrinsic
incompleteness in the record is problematic but should not prevent reasonable assertions on GLOF
triggers to be made, especially if global-scale and consistent patterns in GLOF behaviour are observed.
Future research should therefore more systematically study the factors influencing GLOF frequency and
magnitude and lake formation where a distinction between GLOF conditioning and triggering factors will
be helpful (e.g. Gardelle et al. 2011).
If climate (such as temperature time series) influences GLOFs, as surely must be the case, long lag times
are necessarily implied by the empirical datasets.  With such lags as we have modeled, this brings the
increase of GLOFs following 1930 into line with temperature increases at the end of the Little Ice Age.
Subsequent changes in the GLOF rate (including a several decades-long fall in GLOF rates) similarly can
be attributed to fluctuations in global warming.  If these conclusions are broadly correct, a further
implication is that an acceleration in GLOF rates will probably occur in the 21[st] century, perhaps starting
rather soon. Even though the actual global warming rate for the 21$^{st}$ century may be nearly constant, as
modelled, the fitted warming rate as plotted in Figure 1 panel F accelerates because of memory of post-
LIA, pre-Anthropogenic quasi-stable climate; we are entering a stage where Anthropogenic warming will
increasingly dominate GLOF activity and attribution of GLOFs to Anthropogenic Global Warming will be
confirmed.  For now, this remains a hypothetical projection or expectation and is not yet borne out in
the GLOF record.

**5.   Conclusions**
We conclude that the global record of GLOF following failure of moraine dams  shows a dramatic
increase in GLOF occurrences from 1930 to 1970, then a decline.  We also observe that the GLOF
frequency has not fluctuated directly in response to global climate.  A reasonable premise is that
climate, glaciers, glacier lakes, and GLOFs are closely connected, but the connections between climate
and GLOFs is hidden in response time dynamics.   We argue that response times do not necessarily
reflect linear processes and that lake growth may result in none, single or multiple GLOFs from the same
lake systems. Accordingly, the response times must vary widely from region to region and glacier to
glacier. From this we infer that the 1930 to 1970 increase in global GLOF activity is likely a delayed
response following warming that ended the LIA and decreased rate of moraine dammed lake formation.
We also infer that the decrease in GLOF frequency after 1970 is likely related to a delayed response to
the stabilization of climate following the LIA. In addition, a minor cause (though important locally, for
instance in Peru and Switzerland in particular), GLOF mitigation engineering may have circumvented a
few GLOFs, thus contributing to the downward trend in recent decades.  We can expect a substantial
increase in GLOF incidence throughout the 21st century as glaciers and lakes respond more dynamically
to anthropogenic climate warming.  This is corroborated by recent modelling studies projecting the
location, number and dimension of new lakes in areas where glacier will recede over the coming
decades in the Alps, the Himalayas or the Andes (Linsbauer et al. 2016).

As a result, we argue that the sharply increased GLOF rates starting from 1930 followed by reduced
GLOF frequency from high levels in the mid-20$^{th}$ century are both real and we speculate these trends
may reflect the failure of sensitive glacial lake systems in a lagged response to initial glacier recession
from LIA limits.  The apparent robustness of contemporary lake systems suggests that only the most
resilient moraine-dammed lakes have survived recent climate change. Predicting their future behaviour
is of great importance for those living and working in mountain communities and those developing and
planning infrastructure in such regions.

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

Met Office Hadley Centre Climate Programme (GA01101).  John Kennedy of the Met Office Hadley
Centre provided advice on handling the temperature observation datasets used in this project.
Contributions by JSK, UKH, DHS, and DR were supported by NASA's Understanding Changes in High
Mountain Asia program, the NASA/USAID SERVIR Applied Science Team program, and by the United
Nations Development Program.  We thank C Scott Watson and an anonymous reviewer for their detailed
and incisive reviews of the paper.  We also thank Georg Veh, Jonathan Carrivik and Sergey
Chernomorets  for further comments and clarifications of the inventory.

**Author contributions**

The project was designed by SH following discussion with JK, CH and JR.  Climate model data were
provided by AW and RAB.  Data analysis was carried out by SH, JK, DHS, LR and UH.  JR, VV and AE
provided inventory data.  All authors helped write and review the text.

**Competing Financial Interests**

There are no competing financial interests.

**Figures**

**Figure 1.** Reconciliation of GLOF and climate records.  (**A**) Blue curve: Composite record of northern
hemisphere land surface temperature (merged from multi-proxy data and instrumental records, as
described in the main text), plus a model of land surface temperature during the period 2015-2100.
Red, grey, and black curves: Moving historical averages of the blue curve, as described in the text, using
$\tau_{GLOF}$ = 20, 40, and 80 years, respectively.  (**B** and **C**) Zoom to the more recent periods covered in panel
A.  (**D**) Warming rate extracted from the moving historical averages using $\tau_{GLOF}$ = 20, 40, and 80 years.
Periods of cooling and warming are shown with blue and red tints, respectively, using the $\tau_{GLOF}$ = 80
years curve. (**E**) Zoom-in of panel **D** to a more recent period. (**F**) Comparison of a smoothed GLOF
frequency curve (red line, 10-year historical moving average) with the moving historical average
northern hemisphere temperature (black curve) using $\tau_{GLOF}$ = 80 years and shifted +45 years, where the
45-year shift is considered to be reflective of $\tau_t$, the GLOF trigger timescale.  See supplement text for
more description and explanation.

**Figure 2A-F (Left)**: Temporal distribution of regional GLOF frequency and magnitude.  At all locations,
the cumulative sum of events (black line) indicates an upsurge in the number of events per year. The
timing of this upsurge differs by location and likely reflects an increase in reporting, especially in the
early part of the record, rather than a change in GLOFs, at least until the 1970-90s after which the GLOF
rate reduces. **(Right)** Global time series climate data from the five regions using: CRUTEM 4.2; NOAA
NCDC; NASA GISTEMP. Grey columns represent the baseline against which temperature is measured.
**Figure 3A.**  Record of all precisely dated GLOFs from 1860-2011. (**B**) Wavelet power spectrum of global
GLOF record, significant at 5%. (**C**) Frequency-integrated wavelet power spectrum.
**Figure 4** Seasonal variation in occurrence of GLOF associated with failure of moraine dams.   Only a
proportion of the GLOFs have seasonal data on timing.
**Figure 5.** Temperature anomalies in the CRUTEM4.2 dataset for each mountain region. For each region
we extract all the gridpoints that contain a glacier as defined in the Extended World Glacier Inventory
(WGI-XF) and these are shown as black crosses.

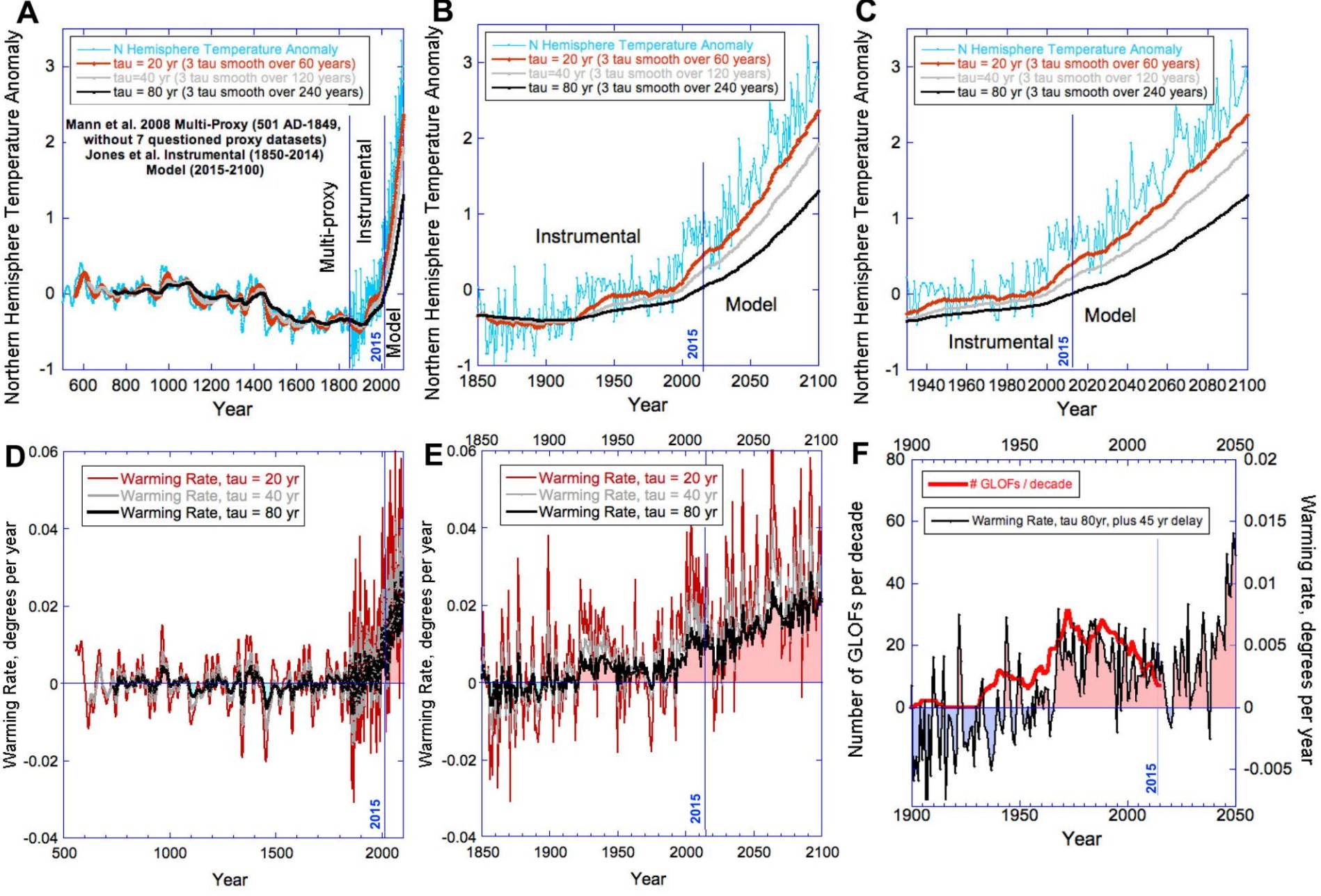

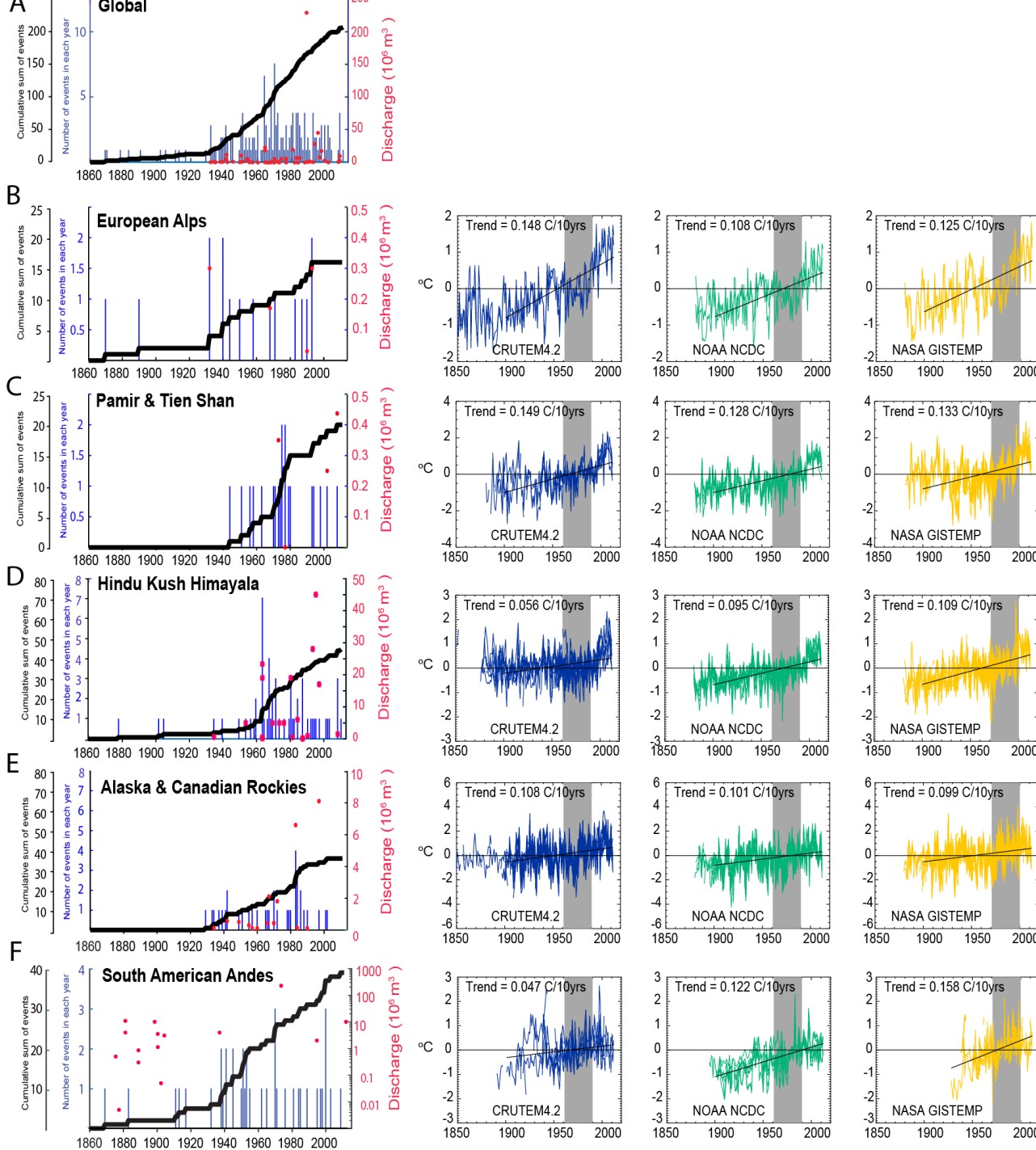

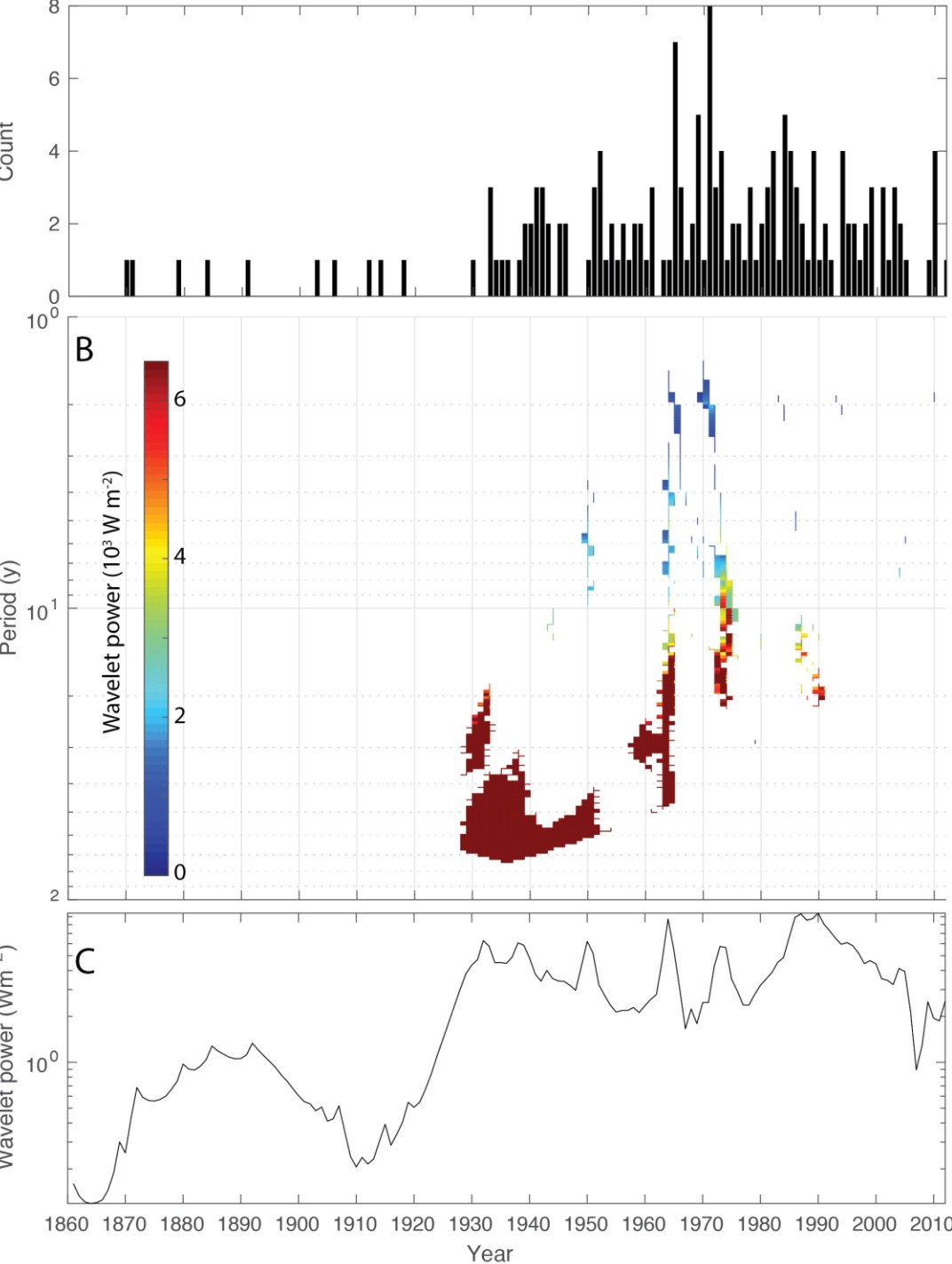

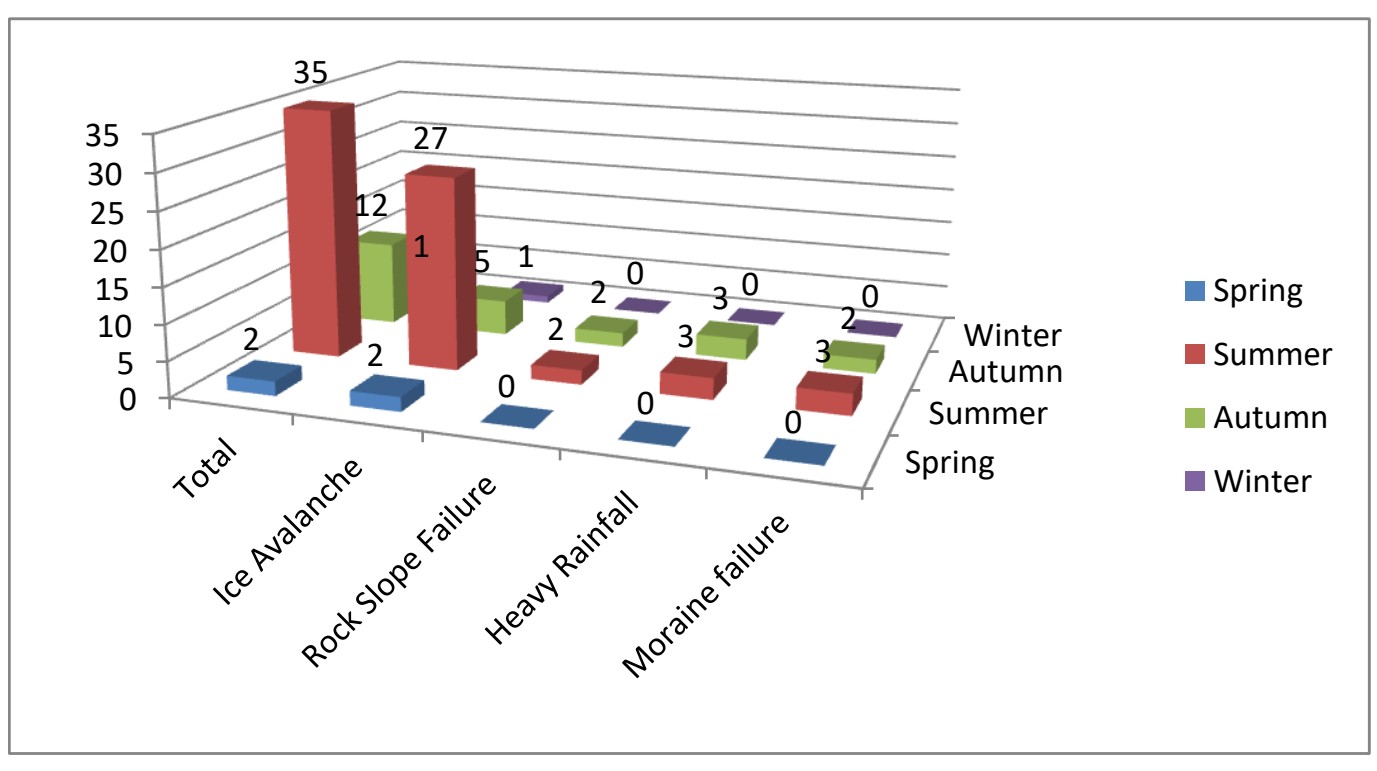

## European Alps

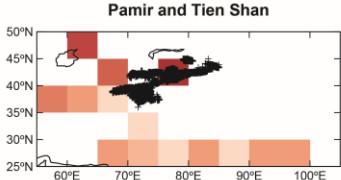

## Pamir and Tien Shan

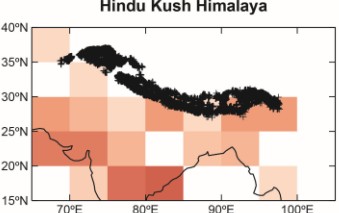

## Hindu Kush Himalaya

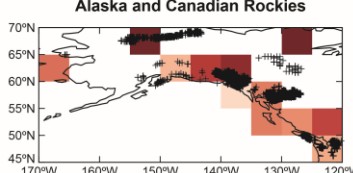

## Alaska and Canadian Rockies

## South American Andes

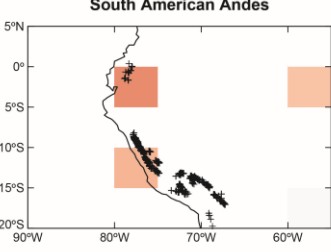

CRUTEM4.2
1991-2012 relative to 1901-1920

**Temperature °C**

0   0.3   0.6   0.9   1.2   1.5   1.8   2.1