# Peer review of "Title: Climate change and the global pattern of moraine-dammed glacial lake outburst floods"

_The Cryosphere, 2017_

## Referee Comment (RC1) · Anonymous Referee #1 · 4 Dec 2017

This is an interesting and timely study on the frequency of glacier lake outbursts from moraine failures. The main findings are an increase of such events around the 1930s and a decrease in recent decades. The study should certainly be published, but I recommend consideration of my below comments:

(1) Methods 2 introduces a model over several pages, but at the end the model is "just" used to smooth the temperature time series, if I understood correctly. Is this long model intro really needed? Wouldn't some running mean filter or similar over a reasonable time span give very different results and provide different explanations to the LIA-1930 lake outburst lag time? If you really find your model is essential, and simpler forms of smoothing don't work I recommend you explain that better and take up the model again in the discussion and conclusions. As said above, I think the most important results

[Figure]

are increase and decrease of outbursts, and you say so, too. I cannot see how this conclusion should depend so much on the temperature time series analysis. If, I am wrong, please explain better.

(2) You need to discuss more what type of processes your model is able to describe what not. Moraine lake failures can be quite different in different regions, for instance regarding ground thermal conditions and possible influence of ground ice and permafrost; topography; glacial history; etc. I think, we would need to understand the geomorphological time scales involved in lake evolution and failure better to better design and understand statistical analyses like yours. I am not saying you have to do that, but you should better discuss that including references to these differences.

(3) Your result to expect a new increase of moraine lake outburst in the future, after a lag time to current atmospheric warming, assumes a constant system status also in the future. I am not so sure this is actually true, in particular not for the mountain cryosphere. If the conditions change into a different system status your extrapolation doesn't hold. A good example for that are thermokarst processes, which are actually involved in the evolution of most glacier lakes. After having been initiated (likely through a rise in temperature, true) they continue to develop even under constant temperatures. In other words, once you have thermokarst processes running, they will continue to increase lakes almost independent of atmospheric temperatures, unless you cool down so much that glaciers grow again significantly. In this example, your extrapolation holds only if the recent acceleration in temperature increase initiates new thermokarst processes. There might also be other positive feedback processes involved in lake growth and outburst that don't require an increase in temperature. Another argument why your assumed constant system status could perhaps not hold are the glaciers themselves; they are in a very different status than after LIA.

(4) Could the 1930s increase of outbursts be related to an improvement of communication capabilities?

(5) Your main finding of recent decrease in outburst numbers agrees with Carrivick and Tweed (2016). You should mention that, and perhaps also else compare your main findings with them.

(6) You acknowledge that preventive measures could have reduced the outburst number in recent decades. Hopefully! You could try to quantify this as most of these measures should be known (and you have a co-author consortium that will know).

(7) Line 376: Again, this assumes somehow similar geomorphological processes and time scales over all regions (see above).

(8) Line 377: From what I understand from your study, a good way to supplement/correlate it would be to check the development of lake areas and numbers. This is much easier (from satellite images, for instance) than outburst statistics. There are such studies, e.g. Gardelle et al. (2011) https://doi.org/10.1016/j.gloplacha.2010.10.003

(9) Much of your data comes likely from other inventories. However, these are not referenced in the main text nor else acknowledged, besides one mention. What would you say if others use in the future your refined database without referencing your paper?

(10) Besides the database itself, I think most, including references, of the Supplement should actually go to the main text, or at least in an Appendix. Some important explanations are too much hidden in the Supplement and not really supplementary information.

(11) Fig.2: are the many temperature trends really necessary for your main messages? Supplement?

(12) 'Methods 1' and 'Methods 2' are not a section numbering according to TC convention: 2.1. , 2.2, etc.

(13) At a few occasions it might be necessary to adapt to the TC style, please check the TC instructions.

[Figure]

(14) There are a few typos and small grammar errors spread over the manuscript.

end

---

## Referee Comment (RC2) · C. S. Watson (Referee) · 11 Dec 2017

Harrison et al. suggest that an observed increase in glacial lake outburst floods from moraine-dammed lakes beginning around 1930 is in response to post-Little Ice Age warming. The authors therefore predict increased GLOF frequencies in coming decades in response to anthropogenic climate warming. The study is of wide and significant interest and is a valuable compilation of data that would be well received in this field. My main concern is that the paper contains contradictory statements regarding the observed increase in GLOFs and the role of reporting bias (specific comment 6), which is acknowledged as a problem but also dismissed without detail of any investigatory analysis. The bias requires more attention in order to justify the conclusions made by the paper.

**Specific comments**

1.  Climate change is mentioned numerous times in the introduction, but the reader doesn't get a sense of what aspects of climate change are important in the context of glacier thinning/retreat leading specifically the formation of moraine dammed lakes. Additionally, what about critical stages in lake formation whereby lake development can proceed independently of warming temperatures. Be more specific about the type of glaciers susceptible to lake development and provide details of the lake evolution process. Projections of increased GLOF frequency in the future can then be grounded in this literature.

2.  I was surprised not to see comparisons made with Carrivick et al. (2016) who also observed a reduction in the number of glacier floods in recent decades (although not exclusively considering moraine-dammed lakes).

3.  L121. Suggest changing to: '…which can lead to moraine failure…' because it's not inevitable.

4.  L146 State how many events were not considered based on this filter.

5.  L225. Please add citations supporting a 20 year lake growth period.

6.  There could be no significant change in GLOF frequency across the whole study period and the changes observed be simply down to observation bias. There are contradicting statements to this effect, which require clarification in the paper: L354-360 It is stated that GLOF frequency increased dramatically and significantly around 1930 globally and 1930-60 regionally, and that there was 'no obvious reason for an abrupt improvement in GLOF reporting in 1930'. However, the incompleteness of the record is then acknowledged as a 'pervasive factor throughout the early period'. L650-653 it is stated that the upsurge in GLOF events per year (which is spatially variable) likely reflects 'an increase in reporting, especially in the early part of the record, rather than a change in GLOFs, at least until the 1970-90s after which the GLOF rate reduces.' While it's stated that you find no obvious reason for the abrupt improvement in reporting, no detail about this analysis is given. Since the whole premise of the study is based on a change in GLOF frequency, you need to be confident there is no reporting bias in the results presented, or that if there is (as would be assumed) how it was investigated and considered throughout the study. At the moment it's not clear how it affects the results and conclusions and that the trends observed are due to post-LIA warming, rather than a bias, or combination of both (and

if so the contributions of each). While contemporary reporting is complete in some regions (European Alps), there are still likely notable omissions in parts of the Himalaya.

**Technical corrections**

L74 '…consequence of climate change…'? To be consistent throughout.

L80 'Carrivick' – check throughout

L128-130 Commas required here and in some other places.

L145-146 This sentence is just a repeat of the first.

L147 '…and attributed to moraine dam failure…'

Regions where moraine dammed lakes are found to form?
L165 '…Alps, no…'

L402 'supraglacial ponds'?

Figure 4 Missing y-axis label. State the proportion of GLOFs with timing information.

**References**

Carrivick, J.L. and Tweed, F.S. 2016. A global assessment of the societal impacts of glacier outburst floods. *Global and Planetary Change.* **144**, 1-16.

---

## Editor Comment (EC1) · C. R. Stokes (Editor) · 5 Jan 2018

I would like to thank the reviewers for their comments on this manuscript. It would appear that both are generally supportive of publication of this important research, but both raise some issues that require careful consideration. If the authors feel that they can address these issues then I would certainly consider a revised manuscript.

---

## Author Response (AR1)

[revised manuscript text omitted]

THIS>>>>>>>>>>>>>>>>>>>>>>>>>>>>>>>>>>>>>>>>The reviewer makes some very important points here
and these go to the heart of the problems of employing simple attribution studies on natural hazards
associated with climate change.  We argue agree that while the original driver of lake development is
likely to involve climate change (resulting in glacier downwasting and slowed  meltwater flux through
glaciers systems as glacier surfaces reduce in gradient) other mechanical and thermodynamic processes
likely assume more importance as the lakes evolve and these includes small-scale calving and insolation-
induced melting of ice cliffs (REFSe.g. Watson et al., 2017).  In our revised manuscript we will discuss
these factors and the ways in which glacial lakes might evolve in the future independently of climate and
the implications for GLOFs that apply.
We also recognise that contemporary mountain glaciers are dissimilar to those that existed in the LIA.
They are, in the main, shorter, thinner and with prominent moraines.  Assumptions that climate
processes acted on similar glacial systems over time are therefore likely to be simplistic. The reviewer
also makes the reasonable point that contemporary glaciers are dissimilar to those in the LIA and
therefore that it would be simplistic to assess these as being similar.  We will make this point too in our
revised manuscript, probably in the expanded discussion section.

Comment [S8]: Refs here

Based on the analysis of our global GLOF database we have shown that a clear trend is detectable
globally and regionally diversified in the 20[th] century with a sharp increase of GLOF occurrence around
1930. This trend is attributable to the observed climate trend, namely the warming since the end of the
LIA. The delayed response of GLOF occurrence is an exemplar model case for the complexities of how
natural systems respond to climate change, underlining the challenges of attribution of climate change
impacts. We have shown here that attribution of GLOFs to climate change is possible although the suite
of factors influencing GLOF occurrence cannot be fully quantified. . REWORD
THIS..............................................In addition, Moraine lake outbursts following moraine failures are likely
to can be quite different in different regions.  This reflects differences in a number of factors including ,
for instance regarding ground thermal conditions, presence or absence of  and possible influences of
ground ice and permafrost, influence of ongoing extreme weather and seismic processes; topography;
and glacial history. To assess these weWe would need to better understand the geomorphological time
scales involved in lake evolution and failure to design a better more robust statistical analysis and to
understand each region's GLOF history.  We thus recommend close attention by the Earth surface process science community to various process time scales using field studies, satellite remote sensing, and theoretical modeling.

Our inventory and the global pattern of GLOFs that is derived from it lacks in many cases precise data on the processes responsible for GLOFs. This is a consequence of incomplete reporting of GLOFs in remote mountain regions, especially before the advent and wide use of remote sensing. In many cases the record is of a large flood being observed and then some time afterwards a collapsed moraine dam is seen and the flood attributed to this collapse. Clearly the precise details of how the collapse occurred is not always available, and this uncertainty bedevils all similar Detection and Attribution studies, especially on those events associated with rapid geomorphological change. This intrinsic incompleteness in the record is problematic but should not prevent reasonable assertions on GLOF triggers to be made, especially if global-scale and consistent patterns in GLOF behaviour are observed. In our revised manuscript we will better discuss the timescales of lake development with reference to the wider literature and also highlight the uncertainties in our approach and the necessity of using inventories that contain uncertainties. Future research should therefore more systematically study the factors influencing GLOF frequency and magnitude and lake formation where a distinction between GLOF conditioning and triggering factors will be helpful (e.g. Gardelle et al. 2011).

REWORD>>>>>>>>>From what I understand from your study, a good way to supplement/ correlate it would be to check the development of lake areas and numbers. This is much easier (from satellite images, for instance) than outburst statistics. There are such studies, e.g. Gardelle et al. (2011) https://doi.org/10.1016/j.gloplacha.2010.10.003

If climate (such as temperature time series) influences GLOFs, as surely must be the case, long lag times are necessarily implied by the empirical datasets. With such lags as we have modeled, this brings the surge increase of GLOFs following 1930 into line with temperature increases at the end of the Little Ice Age. Subsequent changes in the GLOF rate (including a several decades-long fall in GLOF rates) similarly can be attributed to fluctuations in global warming. If these conclusions are broadly correct, a further implication is that an acceleration in GLOF rates will probably occur in the 21$^{st}$ century, perhaps starting rather soon. Even though the actual global warming rate for the 21$^{st}$ century may be nearly constant, as modelled, the fitted warming rate as plotted in Figure 1 panel F accelerates because of memory of post-LIA, pre-Anthropogenic quasi-stable climate; we are getting intoentering a stage where Anthropogenic warming will increasingly dominate GLOF activity and attribution of GLOFs to Anthropogenic Global Warming will be confirmed. For now, this remains a hypothetical projection or expectation and is not yet borne out in the GLOF record.

Reports numerous debris flows (some of them 'glacial debris flows' and GLOFs although not clear whether these are result of failure of moraine dams.

[revised manuscript text omitted]

Dear editor and reviewers
Re: **Climate change and the global pattern of moraine- dammed glacial lake outburst floods**
Manuscript Number: tc-2017-203
Many thanks for your helpful, critical and constructive comments, which significantly helped to
improve our manuscript. Below you will find our detailed answers (in red) to all reviewer comments.
With our best regards.
Stephan Harrison (on behalf of the co-authors).
Comments from Referee 1
This is an interesting and timely study on the frequency of glacier lake outbursts from
moraine failures. The main findings are an increase of such events around the 1930s
and a decrease in recent decades. The study should certainly be published, but I
recommend consideration of my below comments:
(1) Methods 2 introduces a model over several pages, but at the end the model is "just"
used to smooth the temperature time series, if I understood correctly. Is this long model
intro really needed? Wouldn't some running mean filter or similar over a reasonable
time span give very different results and provide different explanations to the LIA-1930
lake outburst lag time? If you really find your model is essential, and simpler forms of
smoothing don't work I recommend you explain that better and take up the model again
in the discussion and conclusions. As said above, I think the most important results are increase and
decrease of outbursts, and you say so, too. I cannot see how this conclusion should depend so much on
the temperature time series analysis. If, I am wrong, please explain better.
Authors' response:
We thank the reviewer for their supporting comments on our paper.  We have considered the length of
the model introduction and have made many small edits that together tighten this section and clarify its
purpose, making a net reduction in its length by around 10%.  It was difficult to reduce it much more
because the type of smoothing we did is novel and there was a need to clarify its purpose.  The reviewer
is correct to say that we have smoothed the data.  However, we did this in order to show how we might
expect GLOFs to have evolved in the past given different glacier and lake response times to climate
forcing.  We agree that the most important result is the finding that the frequency of GLOFs has
decreased in recent decades even as glaciers have continued to melt.  We used the temperature time
series to demonstrate warming in all major mountain regions and to highlight the complexity of the
relationship between glacier hazards and climate change.  This should make us question the simple
causality in Detection and Attribution programmes when dealing with geological hazards which are
assumed to be driven by climate change.
Response in the paper: In our revised version of the manuscript we have more clearly explained the
reasoning behind our use of the model and also highlight the issues for Detection and Attribution
studies that our analysis produces.  We more clearly discuss the model in section 2 under Methods 2.

(2) You need to discuss more what type of processes your model is able to describe
what not. Moraine lake failures can be quite different in different regions, for instance
regarding ground thermal conditions and possible influence of ground ice and permafrost;
topography; glacial history; etc. I think, we would need to understand the
geomorphological time scales involved in lake evolution and failure better to better design
and understand statistical analyses like yours. I am not saying you have to do that,
but you should better discuss that including references to these differences.
Authors' response:
The reviewer is correct to point out the various ways in which GLOFs could occur in different regions and
the range of processes that might be responsible. Our findings regarding the GLOF record's abrupt
increase, peaking, then decrease in GLOF frequency, and our attempt to make a mathematical
smoothing with a retrospective filter clearly is not the end of a search for attribution. We have merely
shown that a climate change attribution incorporating the end of the Little Ice Age is a plausible cause.
It begs for further examination of glacier and limnological response times. Moraine lake failures can be
quite different in different regions, for instance regarding ground thermal conditions and possible
influences of ground ice and permafrost, ongoing extreme weather and seismic processes; topography;
and glacial history. We would need to understand the geomorphological time scales involved in lake
evolution and failure to design a better statistical analysis and to understand each region's GLOF history.
We thus recommend close attention by the Earth surface process science community to various process
time scales using field studies, satellite remote sensing, and theoretical modeling.
Our inventory and the global pattern of GLOFs that is derived from it lacks in many cases precise data on
the processes responsible for GLOFs. This is a consequence of incomplete reporting of GLOFs in remote
mountain regions, especially before the advent and wide use of remote sensing. In many cases the
record is of a large flood being observed and then some time afterwards a collapsed moraine dam is
seen and the flood attributed to this collapse. Clearly the precise details of how the collapse occurred is
not always available, and this uncertainty bedevils all similar Detection and Attribution studies,
especially on those events associated with rapid geomorphological change. This intrinsic
incompleteness in the record is problematic but should not prevent reasonable assertions on GLOF
triggers to be made, especially if global-scale and consistent patterns in GLOF behaviour are observed.
In our revised manuscript we will better discuss the timescales of lake development with reference to
the wider literature and also highlight the uncertainties in our approach and the necessity of using
inventories that contain uncertainties.
Response in the paper: we have discussed time lags in lake development following glacier recession. We
have added references to this (including timescales related to destabilization of mountain slopes
producing mass movements into lakes. This represents the period of paraglaciation (e.g. Ballantyne
2002; Holm et al. 2004; Knight and Harrison 2013).
(3) Your result to expect a new increase of moraine lake outburst in the future, after
a lag time to current atmospheric warming, assumes a constant system status also
in the future. I am not so sure this is actually true, in particular not for the mountain
cryosphere. If the conditions change into a different system status your extrapolation
doesn't hold. A good example for that are thermokarst processes, which are actually
involved in the evolution of most glacier lakes. After having been initiated (likely through
a rise in temperature, true) they continue to develop even under constant temperatures.
In other words, once you have thermokarst processes running, they will continue to increase lakes almost independent of atmospheric temperatures, unless you cool down
so much that glaciers grow again significantly. In this example, your extrapolation holds
only if the recent acceleration in temperature increase initiates new thermokarst processes.
There might also be other positive feedback processes involved in lake growth
and outburst that don't require an increase in temperature. Another argument why your
assumed constant system status could perhaps not hold are the glaciers themselves;
they are in a very different status than after LIA.
Authors' response:
The reviewer makes some very important points here and these go to the heart of the problems of
employing simple attribution studies on natural hazards associated with climate change.  We agree that
while the original driver of lake development is likely to involve climate change (resulting in glacier
downwasting and slowed  meltwater flux through glaciers systems as glacier surfaces reduce in gradient)
other mechanical and thermodynamic processes likely assume more importance as the lakes evolve and
these includes small-scale calving and insolation-induced melting of ice cliffs.  In our revised manuscript
we will discuss these factors and the ways in which glacial lakes might evolve in the future
independently of climate and the implications for GLOFs that apply.
Response in the paper: The reviewer also makes the reasonable point that contemporary glaciers are
dissimilar to those in the LIA and therefore that it would be simplistic to assess these as being similar.
We have made this point in our revised manuscript, probably in the discussion section.
(4) Could the 1930s increase of outbursts be related to an improvement of communication capabilities?
Authors' response
We discuss this issue in Lines 354-360.  Our view is that the widespread nature of this increase is most
likely not a result of increased communication and probably reflects a real change in the data.  This is
most likely to be true for regions with well-developed infrastructure at that time and where glaciers and
human infrastructure are spatially closely linked (such as the European Alps).
Response in the paper: We have further discussed this point in the Discussions section
(5) Your main finding of recent decrease in outburst numbers agrees with Carrivick and
Tweed (2016). You should mention that, and perhaps also else compare your main
findings with them.
Authors' response:
We refer to Carrivick and Tweed (2016) several times in our paper.  However, in the light of the
reviewer's comment in our revised version we will discuss their results in the context of the reduction in
GLOFs.  In their paper they put forward several reasons why 'glacial floods' may have decreased in
frequency in recent decades.  These included successful efforts to stabilize moraine dams and changes in
the ability of fluvial systems to transmit floods over time.  We argue, conversely, that this reduction may
represent a 'lagged' response to glacier perturbations following a climate change.  More research is
clearly needed on this question, and we believe that our analysis, along with that of Carrivick and
Tweed's, will stimulate further work and discussion.
Response in the paper: We have added a section saying this.
(6) You acknowledge that preventive measures could have reduced the outburst number
in recent decades. Hopefully! You could try to quantify this as most of these
measures should be known (and you have a co-author consortium that will know).
Authors' response

Yes, we can provide information to quantify the effects of this remediation on glacier lakes. This has been particularly important in Peru but has also been carried out in the Himalayas (especially in Nepal). Several of the authors have published extensively on these issues and we will address this in a revised manuscript.
Response in the paper: We have added a reference to support this.

(7) Line 376: Again, this assumes somehow similar geomorphological processes and time scales over all regions (see above).
Authors' response
Yes, this is true. However, the global scale of analysis we have adopted means that we are unable to assess the role of variations in geomorphological processes in producing changes in GLOF frequency. Despite this, we will discuss these ideas in a new section describing the uncertainties in our analysis.
Response in the paper: we have discussed this in the revised paper.

(8) Line 377: From what I understand from your study, a good way to supplement/ correlate it would be to check the development of lake areas and numbers. This is much easier (from satellite images, for instance) than outburst statistics. There are such studies, e.g. Gardelle et al. (2011) https://doi.org/10.1016/j.gloplacha.2010.10.003
Authors' response
We agree with this suggestion and will discuss this in the context of future analyses and cite the work of Julie Gardelle and colleagues. However, for us to achieve analysis of this at a global scale would be extremely difficult and would take the paper beyond its original focus. However, one of the co-authors (Adam Emmer) has worked on these issues and can provide better context in a revised manuscript.
Response in the paper: We have made this point and cited Gardelle et al. 2011.

(9) Much of your data comes likely from other inventories. However, these are not referenced in the main text nor else acknowledged, besides one mention. What would you say if others use in the future your refined database without referencing your paper?
Authors' response
We will reference the various inventories we used (and referenced in the Supplementary Information file) in the main manuscript in our revised version.
Response in the paper: We have added references.

(10) Besides the database itself, I think most, including references, of the Supplement should actually go to the main text, or at least in an Appendix. Some important explanations are too much hidden in the Supplement and not really supplementary information.
Authors' response
We will review the information in the Supplementary File and make changes when we think analysis or discussion should be in the main paper.
Response in the paper: We have added this.

(11) Fig.2: are the many temperature trends really necessary for your main messages? Supplement?
Authors' response

We used these temperature trends to show that all the main mountain regions are undergoing
considerable warming; this sets the context for assessing the relationship between GLOFs and climate
warming as a driver.
(12) 'Methods 1' and 'Methods 2' are not a section numbering according to TC convention:
2.1. , 2.2, etc.
Authors' response
OK, we will revise this in the final manuscript.
Response in the paper: We have changed this.
(13) At a few occasions it might be necessary to adapt to the TC style, please check
the TC instructions.
Authors' response
OK.
Response in the paper: Done.
(14) There are a few typos and small grammar errors spread over the manuscript.
Authors' response
We will make sure that these errors are omitted in the final manuscript.
Response in the paper: We have changed these.

Comments from Referee 2

Harrison et al. suggest that an observed increase in glacial lake outburst floods from moraine-dammed
lakes beginning around 1930 is in response to post-Little Ice Age warming. The authors therefore predict
increased GLOF frequencies in coming decades in response to anthropogenic climate warming. The
study is of wide and significant interest and is a valuable compilation of data that would be well received
in this field. My main concern is that the paper contains contradictory statements regarding the
observed increase in GLOFs and the role of reporting bias (specific comment 6), which is acknowledged
as a problem but also dismissed without detail of any investigatory analysis. The bias requires more
attention in order to justify the conclusions made by the paper.
Authors' response
We thank the reviewer for their supportive comments and suggestions for improvement.
Response in the paper: we have discussed this issue more fully in the revised paper.
**Specific comments**
1. Climate change is mentioned numerous times in the introduction, but the reader doesn't get a sense
of what aspects of climate change are important in the context of glacier thinning/retreat leading
specifically the formation of moraine dammed lakes. Additionally, what about critical stages in lake
formation whereby lake development can proceed independently of warming temperatures. Be more
specific about the type of glaciers susceptible to lake development and provide details of the lake
evolution process. Projections of increased GLOF frequency in the future can then be grounded in this
literature.
Authors' response

The reviewer makes some useful suggestions here. In addition to regional warming, other climate change parameters that may lead to glacier lake growth can be changes in wind, humidity, cloud cover, and precipitation. We make an assumption that these are all connected to global or hemispheric warming. In any case, for simplicity of conveying our basic idea that recent elapsed climate change underlies the growth of glacial lakes and the GLOF record, we consider simply the climate record of a geographically broad (northern hemisphere) measure of warming.

We therefore agree that the term 'climate change' masks a wide range of climate processes that drive different geomorphological processes. This means that we must examine more closely the issue of glacier and lake development in response to climate forcing. To do this we must accept that the global pattern of GLOFs and glacier recession over varying timescales may integrate climate processes but also that for detailed (regional or local spatial scales) then glacier hypsometry and glacier type is likely to be a strong control on lake and eventual GLOF evolution. In our revised manuscript we will discuss this with reference to the wider literature (see our response to comment 3 by Referee 1).

Response in the paper: We have added this.

2. I was surprised not to see comparisons made with Carrivick et al. (2016) who also observed a reduction in the number of glacier floods in recent decades (although not exclusively considering moraine-dammed lakes).

Authors' response.
We agree. We did cite Carrivick and Tweed (2016) but agree that we should also discuss their work in the context of the reduction of GLOFs in recent decades. See also our response to comment 5 by Referee 1.

Response in the paper: We have discussed their work in more detail.

3. L121. Suggest changing to: '...which can lead to moraine failure...' because it's not inevitable.

Authors' response
Agree. We will make this change.

Response in the paper: done.

4. L146 State how many events were not considered based on this filter.

Authors' response
OK. We will do this.

Response in the paper: done.

5. L225. Please add citations supporting a 20 year lake growth period.

Authors' response
Yes, we will do this. There are numerous examples from temperate regions where glacier lakes have developed over the past 20 years or so and we will cite these.

Response in the paper: done.

6. There could be no significant change in GLOF frequency across the whole study period and the
changes observed be simply down to observation bias. There are contradicting statements to this effect,
which require clarification in the paper:
Response in the paper: we have further discussed this in the Discussion section.

L354-360 It is stated that GLOF frequency increased dramatically and significantly around 1930 globally
and 1930-60 regionally, and that there was 'no obvious reason for an abrupt improvement in GLOF
reporting in 1930'. However, the incompleteness of the record is then acknowledged as a 'pervasive
factor throughout the early period'. L650-653 it is stated that the upsurge in GLOF events per year
(which is spatially variable) likely reflects 'an increase in reporting, especially in the early part of the
record, rather than a change in GLOFs, at least until the 1970-90s after which the GLOF rate reduces.'
While it's stated that you find no obvious reason for the abrupt improvement in reporting, no detail
about this analysis is given. Since the whole premise of the study is based on a change in GLOF
frequency, you need to be confident there is no reporting bias in the results presented, or that if there is
(as would be assumed) how it was investigated and considered throughout the study. At the moment
it's not clear how it affects the results and conclusions and that the trends observed are due to post-LIA
warming, rather than a bias, or combination of both (andif so the contributions of each). While
contemporary reporting is complete in some regions (European Alps), there are still likely notable
omissions in parts of the Himalaya.
Authors' response
The issue of reporting bias is a very difficult one and we address this in the paper.  However, we are
confident in our main conclusion (that GLOFs have decreased in recent decades) and also confident that
this cannot be a consequence of reporting bias.  We will clarify this argument in the discussion section of
the revised manuscript. We also accept that there may be incomplete contemporary reporting in some
remote regions, such as parts of the Himalayas.  However, we would stress that this is one of the first
attempts to analyse GLOF frequencies at a global scale and we also accept that this paper will not be the
last word on this issue, but is likely to stimulate further research.

Response in the paper: we have added to this debate.

**Technical corrections**
L74 '…consequence of climate change…'? To be consistent throughout.
Authors' response
We agree and will be consistent in the use of this term in the revised manuscript.
Response in the paper: done.

L80 'Carrivick' – check throughout
Authors' response
We will make sure that this spelling is correct.
Response in the paper: done.

L128-130 Commas required here and in some other places.

Authors' response
OK.
Response in the paper: done.

L145-146 This sentence is just a repeat of the first.
Authors' response
Agreed.  We will omit this.

L147 '…and attributed to moraine dam failure…'
Regions where moraine dammed lakes are found to form? L165 '…Alps, no…'
Authors' response
NOT SURE WHAT THE REVIEWER IS SUGGESTING HERE

L402 'supraglacial ponds'?
Authors' response
Yes we agree and will change this.
Response in the paper: done.

Figure 4 Missing y-axis label. State the proportion of GLOFs with timing information.
Authors' response
We will make these additions.